# Immune Checkpoint Inhibitors in Esophageal Cancers: Are We Finally Finding the Right Path in the Mist?

**DOI:** 10.3390/ijms21051658

**Published:** 2020-02-28

**Authors:** Caterina Vivaldi, Silvia Catanese, Valentina Massa, Irene Pecora, Francesca Salani, Stefano Santi, Monica Lencioni, Enrico Vasile, Alfredo Falcone, Lorenzo Fornaro

**Affiliations:** 1Department of Translational Research and New Technologies in Medicine and Surgery, University of Pisa, Via savi 10, 56126 Pisa PI, Italy; alfredo.falcone@med.unipi.it; 2Unit of Medical Oncology, Pisa University Hospital, Via Roma 67, 56126 Pisa, Italy; catanesesilvia@gmail.com (S.C.); valentinamassa22@gmail.com (V.M.); irene.pecora@gmail.com (I.P.); f.salani1@gmail.com (F.S.); m.lencioni@ao-pisa.toscana.it (M.L.); envasile@gmail.com (E.V.); lorenzo.fornaro@gmail.com (L.F.); 3Esophageal Surgery Unit, Tuscany Regional Referral Center for the Diagnosis and Treatment of Esophageal Disease, Pisa University Hospital, Via Roma 67, 56126 Pisa, Italy; s.santi@ao-pisa.toscana.it

**Keywords:** esophageal cancer, immunotherapy, predictive factors, checkpoint inhibitors

## Abstract

Esophageal cancer remains a challenging disease due to limited treatment options and poor prognosis. In recent years, immune checkpoint inhibitors (ICI) have been proven to be safe and effective in the treatment of highly lethal malignancies, such as non-small cell lung cancer and melanoma. Recent clinical trials also showed promising activity in immune checkpoint inhibitors in pretreated advanced esophageal carcinoma and a potentially significant impact on the outcome of selected patients, independently of histology. Combination studies evaluating immunotherapy and chemotherapy and, in localized disease, radiotherapy are in progress and will hopefully confirm their promises in the near future. However, reliable predictive biomarkers are still lacking. Indeed, at present, the role of programmed cell death ligand 1 expression and other factors (such as microsatellite instability and tumor mutational burden) as predictive biomarkers of benefit to immune checkpoint inhibitors is still controversial. Our aim was to explore the rationale of ICIs in esophageal cancer, review the results already available in multiple settings, and investigate future perspectives with single-agent and combination strategies.

## 1. Introduction

Esophageal cancer is one of the most challenging gastrointestinal tumors. The International Agency for Cancer Research (IARC) estimates it as the ninth most common cancer worldwide and the fifth leading cause of cancer death for the year 2018 [1]. Five year survival rates for all stages are around 20.9% in China and in the USA, and only 12% in Europe [2,3,4]. Standard treatment consists of multidisciplinary management for locoregional and locally advanced disease, and in chemotherapy for palliative treatment of metastatic disease [5], but the survival benefit of available therapies is still very limited.

When talking about esophageal cancer, we necessarily need to look at it not as a single entity. The two major histological subtypes, the squamous cell carcinoma (ESCC) and adenocarcinoma (EAC), are known to differ notably in terms of risk factors and epidemiology. Although ESCC represents about ninety percent of the incident esophageal cancers, especially in Central-Eastern Asia and South America (due to nutritional behavior, tobacco, and alcohol consumption) [6], its incidence had been decreasing in the last decades in western countries. On the contrary, the incidence of EAC has increased, as well as that of the tumors of the proximal region of the stomach and of the cardia, in parallel with obesity, gastro-esophageal reflux disease and Barrett’s esophagus [7].

In 2016, The Cancer Genome Atlas Research Network (TCGA) performed an analysis of 164 esophageal tumors derived from Asian and Caucasian populations with the aim of molecularly separating these two diseases [8]. ESCC revealed a similitude to other squamous cell neoplasms (SCCs) such as head and neck (HNSCC) and squamous non-small cell lung cancer (NSCLC) [9,10]. According to molecular features, ESCC can be classified into three different subtypes: ESCC1, with somatic alterations similar to other SCCs, some of them associated with poor prognosis and resistance to chemo-radiotherapy; ESCC2, with a greater leukocyte infiltration and a higher expression of the bone marrow stromal antigen 2 (BST2) immunomodulatory molecule; and the ESCC3 characterized by alterations predicted to activate the Phosphoinositide 3-kinases (PI3K) pathway and aspects as non-identifiable in other SCCs. These three sub-types seem to have also a distinguished geographic distribution: neoplasm from the Vietnamese population tended to be ESCC1, ESCC2 is more frequent Eastern Europe and South America, and people from Canada and the USA present ESCC3 more often [8]. EAC showed a high prevalence of chromosomal instability (CIN), hence bearing a resemblance to CIN gastric cancer [11]. In this study, 36 adenocarcinomas of the gastro-esophageal junction (GEJ) were additionally evaluated, and it was found an increased incidence of the CIN phenotype moved proximally to the distal esophagus so that the EAC appeared as a chromosomal instability disease. However, some molecular features do not permit the consideration of EAC and proximally gastric tumors as a single entity [8].

Naturally, the identified molecular subgroups contributed to ameliorating our knowledge about the biology of the esophageal cancer disease and offered new potential therapeutic targets. In the last years, multiple target agents have been investigated with quite unsatisfactory results [12].

In a disease with such an awful prognosis and poor therapeutic options, the interest in the impact of immunotherapy is increasing.

Cytotoxic T-lymphocyte antigen 4 (CTLA-4), expressed only on T cells, helps to inhibit the T cell itself and down-modulate the priming phase of the immune response. Programmed cell death protein-1 (PD-1), expressed on the surface of T cells, B cells, monocytes, and natural killer (NK) cells binds PD ligand-1 (PD-L1) and -2 (PD-L2) in order to down-regulate the effector phase of the immune response. Immune-checkpoint inhibitors (addressing PD1, PD-L1, or CTLA-4) break the immune tolerance and restore T cell recognition against tumor cells [13]. Targeting immune checkpoints represents one of the most investigated strategies in oncology with promising results in different tumor types.

In this review, we examine the rationale and the results of clinical trials that could lead to the introduction in the clinical practice of immune checkpoint inhibition in esophageal cancer, with a special focus on predictive and prognostic biomarkers and future perspectives.

## 2. Checkpoint Inhibitors in Advanced Disease

### 2.1. From Initial Phase I-II to Randomized Phase III Trials

Many phase I-II trials have assessed the role of the immune checkpoint blockade in esophageal cancers (Table 1).

#### 2.1.1. KEYNOTE-028 Trial

In the multicohort phase Ib KEYNOTE-028 exploring the efficacy of pembrolizumab in PD-L1-positive solid tumors (PD-L1 positivity defined as membranous staining on at least 1% of scorable cells both neoplastic and contiguous mononuclear inflammatory cells), encouraging results from the esophageal cohort were presented [14]. Of the 23 patients enrolled, around half were Asian. Most patients were male (83%), presented squamous histology (78%), and were previously treated with radiation therapy (61%), and were heavily pre-treated (48% > 3 previous lines of treatment). At a median follow-up of 7 months (range 1-33), overall response rate (ORR) was 30% (95% CI 13-53); moreover, two patients achieved stable disease (SD). An interesting activity was observed in both histology subgroups—28% of ORR for patients with ESCC and 40% for those with EAC, even though the small number does not empower us to confront these results and state any kind of conclusion. A decline in tumor burden from baseline was observed in 52% of patients, and the duration of response (DOR) was 15 months (range 6 to more than 26), with 4 months (range 2-8) as the median time to initial response. Survival data also showed a durable clinical impact, although poor results were found in terms of median progression-free survival (PFS) at 1.8 months (95% CI, 1.7 to 2.9), and the 6 month and 12 month PFS rates were 30% and 22%, respectively. Instead, median overall survival (OS) was 7.0 months (95% CI, 4.3 to 17.7), and the 6 month and 12 month OS rates were 60% and 40%. The safety profile was acceptable—39% of patients experienced adverse events of all grades, and severe adverse events of grade 3 were observed in 17% of patients, which were all resolved with only treatment discontinuation. In this study, a prolonged preliminary clinical benefit was achieved thanks to anti-PD1 blockade in a PD-L1-positive population of esophageal squamous cell carcinoma and adenocarcinoma, although 52% of the trial population experienced progressive disease (PD) as best response. Evidence about the response in the PD-L1-negative population cannot be derived.

#### 2.1.2. KEYNOTE-180 Trial

Emerging data from KEYNOTE-028 provided the rationale for the phase II KEYNOTE-180 trial [15]. Effectiveness of pembrolizumab 200 mg every 3 weeks was assessed in a population of 121 patients affected by advanced or metastatic ESCC (52%) and EAC (48%), previously treated with more than two lines of treatment (12% with three or more lines), with non-Asians amounting to approximately 70% of the entire population. The enrolment was irrespective of PD-L1 expression; 47.9% had PD-L1 positivity, defined as a more than 10% in a combined positive score (CPS) number of PD-L1-positive cells (tumor cells, macrophages, and lymphocytes) divided by the total number of tumor cells, multiplied by 100. After a median follow-up of 13.3 months, activity data confirmed what was previously established in KEYNOTE-028. ORR in the whole population was 9.9% (95% CI, 5.2–16.7), 14.3% (95% CI, 6.7–25.4) among ESCC, and 5.2% (95% CI, 1.1–14.4) among EAC, with an inversed proportion with respect to what was observed in phase Ib regarding to the PD-L1 status, with 13.8% (95% CI, 6.1–25.4) of complete and partial responses being observed among PD-L1-positive tumors and 6.3% (95%CI, 1.8–15.5) among PD-L1-negative tumors. It was also noteworthy that median DOR was not reached among responders at data cut-off analysis. Median PFS was 2 months (95% CI, 1.9–2.1), with a 6 month PFS rate of 16% (95% CI, 10–23) and a 9 month rate PFS of 9% (95% CI, 5–16%), whereas median Overall Survival (mOS) was 5.8 months (95% CI, 4.5–7.2), with a 6 month OS rate of 49% (95% CI, 40–57%) and a 12 month OS rate of 28% (95% CI, 20–37%). These survival data sustain the durability of response irrespective of PD-L1 status in esophageal cancers and are favorably comparable with the literature’s evidence derived from treatment with taxanes [16,17]. Toxicity management was feasible, as only one treatment-related death, due to pneumonitis, was registered.

#### 2.1.3. ATTRACTION-1 Trial

At the same time, the first results with anti-PD-1 nivolumab became available. In 2017, Kudo et al. published the results of the ATTRACTION 1, an open-label single-arm phase II trial, conducted in 65 Japanese patients affected by ESCC refractory or intolerant to standard chemotherapies (one-third of the population had undergone more than four therapy lines) treated with nivolumab 3 mg/kg administered every 2 weeks [18]. After a median follow-up of 10.8 months (interquartile range 4.9–14.3), an overall response rate (ORR) of 17% (95% CI 10–28) by central assessment was achieved, with a lower limit of the confidence interval much above the lower threshold for response expected (5%) from placebo. The centrally assessed disease control rate (DCR) was 42% (95% CI 31–54), with a median duration of OS and PFS of 10.8 months (95% CI 7.4–13.3) and 1.5 months (95% CI 1.4–2.8), respectively. The response was confirmed, even overestimated, when assessed by Immuno-RECIST criteria [19] with respect to RECIST 1.1 criteria. High-grade adverse events were reported in 26% of patients, whereas serious adverse events were reported in 17% of patients. These results showed a manageable safety profile of nivolumab with promising and long-lasting activity in a refractory esophageal cancer Asian population.

#### 2.1.4. KEYNOTE-181 Trial

Moving from results of the abovementioned phase I-II trials, in the last year data emerging from two randomized phase III trials were presented and have definitely shown the role of immunotherapy in esophageal cancer (Table 1). In phase III KEYNOTE-181 trial, the role of pembrolizumab as second-line treatment in advanced esophageal cancer was investigated [20]. A mixed population of 628 patients (38.6% Asian) with a prevalence of squamous histology (63.8%) was randomized to receive pembrolizumab 200 mg every 3 weeks or treatment with paclitaxel, docetaxel, or irinotecan according to the investigator’s choice [16,17,21]. Three co-primary endpoints had to be demonstrated: superiority in terms of OS for the experimental arm in the overall population (intention-to-treat, ITT), in squamous cell tumors, and in tumors with CPS >10%; therefore, the α-spending was strictly designed. Secondary endpoints were ORR, safety, and PFS. The study goals were a statistically significant superiority in terms of survival in the CPS PD-L1 > 10% population for the immunotherapy arm with a mOS of 9.3 months vs. 6.7 months (hazard ratio (HR) 0.69; 95% CI 0.52–0.93; *p* = 0.0074), but also a clinically significant superiority in 12 month OS rate (43% vs. 20%) and 18 month OS rate (26% vs. 11%). In the ESCC population, the statistical significance was not enriched (mOS 8.2 months vs. 7.1 months; HR 0.78; 95% CI 0.63–0.96; *p* = 0.0095), possibly due to the strict statistic design, which might have underestimated a clinical benefit [22], which, on the contrary, could be observed at 1 year (39% vs. 25%) and at 18 month OS rate (23% vs. 12%). In the ITT population, no statistically significant differences in terms of mOS (7.1 months vs. 7.1 months; HR 0.89; 95% CI 0.75–1.05; *p* = 0.0560) were recorded, but a trend for a gain of benefit in the experimental arm might have been perceived at 12 months (32% vs. 24%) and at 18 months (18% vs. 10%). Regarding the histology in the PD-L1-positive population, the benefit in terms of survival derived from pembrolizumab derived benefit in CPS ≥ 10 population seemed to be higher in the ESCC, with a median OS of 10.3 months vs. 6.7 months, whereas mOS was 6.3 months vs. 6.9 months in EAC, although this last component ranked around only 25% of this selected subgroup. For the abovementioned reason, this trial supported pembrolizumab as a new second-line standard of care for esophageal cancer with PD-L1 CPS ≥ 10 and encouraged furthers evaluations of checkpoints inhibitors in ESCC treatment. On July 2019, the Food and Drug Administration (FDA) approved pembrolizumab for patients with recurrent, locally advanced, or metastatic squamous cell carcinoma of the esophagus whose tumors express PD-L1 CPS > 10 %, with disease progression after one or more prior lines of systemic therapy.

#### 2.1.5. ATTRACTION-3 Trial

Analogously, the ATTRACTION-3 trial, a multicentre, randomized phase III trial, compared the anti-PD1 nivolumab to second-line taxanes chemotherapy in patients with refractory and metastatic ESCC [23]. The study was conducted in 419 patients, of which 96% were Asian. After a median follow-up of 17.6 months, more than 76% of events had been realized and no differences in terms of ORR were registered (19% and 22% in the nivolumab and chemotherapy group, respectively), but the duration of response differed between the two groups with a remarkable benefit for immunotherapy compared to chemotherapy (6.9 vs. 3.9 months). Although no benefit in terms of PFS was shown in the experimental arm (HR 1.08, 95% CI 0.87–1-34), with regard to OS, a benefit in favor of the experimental group was demonstrated with an HR of 0.77 (95% CI 0.62–0.96, *p* = 0·0019) after a post-hoc statistic correction due to the presence of non-proportional Kaplan–Meyer curves. Median OS values were 10.9 months (95% CI 9.2–13.3) vs. 8.4 months (95% CI 7.2–9.9), respectively, in the two groups. Notably, the OS curves crossed after 5 months when almost 25% of the patients had died in the nivolumab arm; then, they separated with an 18 month OS rate of 31% vs. 21%. Up against the conclusions of the KEYNOTE-181, no relevant interaction was observed in the pre-specified sub-groups analysis stratified by PD-L1 expression, although there was a difference of 15% between the hazard ratios in favor of nivolumab in the subgroup PD-L1≥1%. This study might establish a new standard of care in the second-line treatment of esophageal squamous cancers; however, because of the high prevalence of Asian people and what was shown about the greater effectiveness of anti-PD1 therapy in this population in the KEYNOTE-181, it cannot be easily extended as a world-wide accepted guideline [22]. Nonetheless, these results definitely encourage promising results of immunotherapy in squamous cell carcinoma.

### 2.2. Innovative Strategies in Advanced Disease: Combination Treatments, and Novel agents

Data from studies conducted up to now suggest that only a subset of patients could benefit from immunotherapy. In addition to trying to identify putative predictive factors, esophageal cancer research is also currently investigating the combination of different immune agents and the combination of immune checkpoint inhibitors (ICIs) with chemotherapy and radiotherapy in order to extend the percentage of patients that could benefit from this novel therapeutic approach (Table 1).

#### 2.2.1. Nivolumab and Ipilimumab Combination

The combination of nivolumab and ipilimumab has demonstrated a consistently impressive anti-tumor activity and durable tumor responses in different solid tumors [24,25,26]. The clinical impact derived from a double blockade of the immunological checkpoints (PD-1 and CTLA-4) in locally advanced or metastatic chemotherapy-refractory gastro-esophageal cancers was explored in the phase I-II, non-comparable, randomized trial CheckMate-032, in which the association between nivolumab and ipilimumab was tested through different schedules [27]. One-hundred and sixty patients were assigned to three different cohorts: nivolumab 3 mg/kg alone (N3), nivolumab 1mg/kg and ipilimumab 3mg/kg (N1 + I3), and nivolumab 3 mg/kg and ipilimumab 1 mg/kg (N3 + I1). In both the combination arms, treatment with doublet was carried out for four cycles, then followed by nivolumab 3mg/kg-only administration. The trial enrolled western patients; only the minority of them (15–17%) were diagnosed with EAC compared to EGJ carcinoma and gastric carcinoma. Around 45% of patients in every cohort had progressed or were intolerant to three or more regimens. Although not powered to assess differences among cohorts, after a median follow-up of 2 years, the best results in terms of activity were observed in the N1+I3 cohort, with an ORR of 12%, 24%, and 8%, and a reduction in tumor burden from baseline of 29%, 45%, and 27% in the N3, N1+I3, and N3+I1 arms, respectively. No relevant differences concerning the median Progression-Free Survival (mPFS) value among the cohorts were registered, but PFS was quite varied between the N1 + I3 cohort (17%) and the other two (8% in the N3 and 10% in the N3 + I1) at 12 months. Conversely, mOS was similar among groups: 6.2 months vs. 6.9 months vs. 4.8 months, in the N3, N1 + I3, N3 + I1, respectively. Despite the improvement in terms of response with the combination of N1+I3 without any advantage in terms of survival, it should be emphasized that this combination was associated with a major incidence of high-grade adverse events of 47% (especially diarrhea and elevation of liver enzymes), compared to 17% of the N3 and 27% of the N3+I1. The potential predictive role of PD-L1 and microsatellite instability (MSI) status was investigated. ORRs seemed superior in PD-L1-positive tumors (N3: 19%, N1 + I3: 40%, N3 + I1: 23%), in comparison to PD-L1-negative tumors (N3: 12%, N1 + I3: 22%, N3 + I1: 0%), similarly to MSI-high tumors (N3: 29%, N1 + I3: 50%, N3 + I1: 50%) compared to MSI-low tumors; however, the slight sample size of PD-L1-positive and MSI-high tumors does not support any conclusion about PD-L1 and MSI status as predictors of response to checkpoint inhibitors.

#### 2.2.2. M7824—Bifunctional Fusion Receptor Protein

In a phase I trial [28], preliminary interesting data were presented for a new drug M7824, a bifunctional fusion protein composed of a human anti-PD-L1 immunoglobin G1 (IgG1) monoclonal antibody (mAb) fused with two extracellular domains of the transforming growth factor beta (TGFβ) receptor II, aiming to magnify the response to anti-PD(L)1 therapy through the inhibition of the TGFβ pathway. As is known, tumor-associated macrophages promote the immunosuppressive environment by acting on T regulatory cells via immunosuppressive cytokines, such as interleukin-10 and TGFβ [29]. Thirty heavily pre-treated Asian patients affected by ESCC presented a confirmed ORR of around 20%. Treatment-related adverse events were mainly cutaneous (13.3% of grade 3), and an interstitial lung disease of all grades was recorded in 10% of cases.

#### 2.2.3. Toripalimab

The humanized immunoglobulin G4 (IgG4) against programmed cell death protein-1 (PD-1), toripalimab (JS001), binds to the FG loop of PD-1 independently of N-linked glycosylation of PD-L1 that would affect the contact between PD-1 and PD-L1 [30]. It was approved by the China National Medical Products Administration in 2018 as a second-line treatment for melanoma, and many phase I-III clinical trials in numerous indications are still ongoing [31]. Attractive data of a phase Ib/II study among 56 Chinese patients with metastatic ESCC, most of them pre-treated with more than two lines of therapy, were presented at the America Society of Clinical Oncology (ASCO) Meeting in 2018. Responses were evaluable in 60% of patients with an ORR of 23.5% and a DCR of 64.7%; remarkably, the entity of response was also evaluated according to PD-L1 status (21.4% defined positive if PD-L1 was present in more than 1% in tumors and immune cells), but no significant variations were found [32].

#### 2.2.4. Combination of Chemotherapy and Immunotherapy

A new horizon of therapeutic strategies has opened after the accumulation of many pieces of evidence showing an improvement in patient outcome through the combination of cytotoxic chemotherapeutic agents and immune checkpoint inhibitors in different lines of treatment, especially in earlier settings in different solid tumors [33,34,35]. In a very small phase Ib trial, dose-limiting toxicity (DLT) of the combination of durvalumab (anti-PDL1 ab) + tremelimumab (anti-CTLA-4) + first-line chemotherapy with 5-fluorouracil and cisplatin was assessed in ESCC [36]. Among the six patients enrolled, the toxicity profile of the combination was acceptable—immune-related adverse events were all low-grade adverse events (rash, pruritus, and diarrhea), whereas high-grade adverse events were all manageable and due to cytotoxic drugs.

## 3. Immunotherapy in Localized Disease

Following promising results in the metastatic setting, immunotherapy is also currently under investigation in localized disease. Although chemo-radiation followed by surgery represents the standard of care in this setting [37], disease relapse occurs in approximately 50% of patients within 1 year after surgery, and the risk of recurrence is significantly higher in patients not achieving complete pathological response [38].

### 3.1. Adjuvant Durvalumab

Moving from this issue and taking into account the preclinical evidence of synergistic activity between chemo-radiation and PD-1 inhibition [39], The Big Ten Cancer Research Consortium recently presented results of an interesting single-arm phase II multicenter trial [40]. The study evaluated adjuvant treatment with the PD-L1 inhibitor durvalumab in patients with locally advanced esophageal or GEJ adenocarcinoma not achieving a pathological complete response after chemoradiation followed by radical resection. Study primary endpoint was 1 year relapse-free survival, and secondary endpoints were safety and feasibility. The authors hypothesized that durvalumab would increase 1 year relapse-free survival to 75% from a historical controlled rate of 50%. Starting from 1 to 3 months after surgery, patients received durvalumab 1500 mg intravenously, every 4 weeks for 1 year. A total of 24 patients resected after carboplatin and paclitaxel (18 patients) or cisplatin and 5-FU (6) and concurrent radiation were enrolled (10 with distal esophageal adenocarcinoma and 14 with GEJ adenocarcinoma). A total of 12 patients completed adjuvant treatment, whereas 12 prematurely stopped treatment (six due to disease recurrence, five AEs, one consent withdrawal). After a median follow up of 14.5 months (range of 1.7-24 months), 17 patients were free of recurrence, and 1 year relapse-free survival (RFS) and OS were 79.2% and 95.5%, respectively. Three patients developed grade 3 immune-related AEs leading to treatment discontinuation (pneumonitis, hepatitis, and colitis). In conclusion, adjuvant durvalumab seems to be feasible and has promising results in patients with residual disease after trimodality therapy for esophageal and GEJ adenocarcinoma and deserves further evaluation in larger trials.

### 3.2. Neoadjuvant Nivolumab

Preclinical evidence suggests that neoadjuvant chemoradiation causes transient upregulation of PD-L1 and other immune checkpoints in esophageal adenocarcinoma, and this mechanism seems to be dose-dependent [41]. Moving from the hypothesis that chemoradiation could enhance immune checkpoint response, Kelly RJ et al. recently presented preliminary data from a phase Ib trial evaluating nivolumab with or without the anti-Lymphocyte-activation gene 3 (LAG3) relatlimab followed by chemoradiation plus nivolumab and surgery in esophageal/GEJ cancer [42]. In the first part of the study, 16 patients were enrolled. Two cycles of nivolumab q14 were administered and followed by concurrent paclitaxel/carboplatin and radiation + nivolumab administered at weeks 1, 3, and 5. Surgery was performed 6-10 weeks after the end of nivolumab administration. The primary endpoints of the study were safety and feasibility. Treatment did not result in unexpected adverse events; grade 3 treatment-related adverse events were reported in four patients but they did not affect the timing and the morbidity of the surgery. A total of 15 out of 16 patients underwent surgery (one patient had metastatic disease before surgery and baseline inconclusive positron emission tomography); nine achieved pathological down-staging including five pathological complete response. Waiting for translational analysis and results of the cohort with nivolumab and anti-LAG3 combination, this study represents an interesting attempt at integrating immunotherapy in preoperative multimodal treatment.

## 4. Ongoing Trials

### 4.1. Pre-Treated Patients

#### 4.1.1. RAMONA Trial

Second-line treatment of the geriatric population, usually poorly represented in clinical trials, is a clear unmet medical need [43]. In this setting, the multicenter open-label phase II RAMONA trial [44] is investigating an immunotherapy approach consisting of a nivolumab monotherapy in conjunction with a safety-guided treatment escalation to a nivolumab and ipilimumab combination regimen in the second-line treatment of elderly ESCC patients (> 65 years) (Table 2). Concerns about related adverse events of combination strategies compared to nivolumab monotherapy were downsized thanks to the results of the CheckMate-012 in advanced NSCLC (10% vs. 13%); however, patients in this study are strictly monitored for the occurrence of treatment-related adverse events. Investigators plan to enroll 75 patients; the primary endpoint is OS, and secondary endpoints are the time to quality of life deterioration, PFS, and ORR.

#### 4.1.2. Toripalimab in Small Cell Esophageal Carcinoma

A phase II trial is exploring the activity of toripalimab in the second-line treatment of small cell esophageal carcinoma. This rare carcinoma presents an immune microenvironment enriched of effector T cells, NK cells, and macrophages with an M2-phenotype that might represent a better milieu to achieve a response to anti-PD1 therapy (NCT03811379) (Table 2).

#### 4.1.3. Tislelizumab in Pretreated Patients

Tislelizumab is a humanized immunoglobulin G4 (IgG4)-variant monoclonal antibody against programmed cell death protein-1 (PD-1) that competitively blocks binding by both PD-L1 and PD-L2, thus enhancing signaling in a T cell. Moreover, it inhibits antibody-dependent cellular cytotoxicity (ADCC), antibody-dependent cellular phagocytosis (ADCP), or complement-dependent cytotoxicity (CDC) effects in humans by binding the gamma fragment crystallizable region (Fc) receptors (FcγR) such as FcγRI and FcγRIIIA expressed on myeloid-derived cells (M2-macrophage, myeloid-derived suppressor cell), and by binding the subunit 1 of the complement 1 [45,46]. The BGB-A317-302 is an ongoing phase III trial comparing tislelizumab (BGB-A317) with chemotherapy chosen by the investigator in metastatic esophageal cancer progression after first-line chemotherapy [47]. OS is the primary endpoint; secondary endpoints include ORR, PFS, DOR, health-related quality of life, and safety data (Table 2).

### 4.2. First-Line Setting

As abovementioned, the association of chemotherapy and immune-checkpoint inhibitors opened fasters routes to reach survival improvement in many solid tumors [33,34,35]. This strategy is under investigation also in the first-line treatment of esophageal cancer (Table 2).

The association of the anti-PD1s, pembrolizumab and nivolumab, with chemotherapy, are enquired into, respectively, in two phase III trials: the KEYNOTE-590 trial [48] and the CheckMate-648 [49].

#### 4.2.1. KEYNOTE-590 Trial

In the randomized, double-blind, placebo-controlled multi-site KEYNOTE-590, a population affected by locally advanced unresectable or metastatic previously untreated ESCC and EAC, or Siewert type 1 adenocarcinoma of the GEJ, was randomized to receive pembrolizumab or placebo in association with platinum and fluorouracil. Primary endpoints were OS and PFS in ITT and in PD-L1 (CPS ≥ 10%) population.

#### 4.2.2. CheckMate-648 Trial

In the CheckMate-648, the combination of nivolumab and ipilimumab vs. nivolumab added to cisplatin and fluorouracil were compared to cisplatin and fluorouracil doublet. Primary endpoints were OS and PFS by central assessment in patients with a PD-L1 positive tumor (defined as PD-L1 expression on ≥ 1% of tumor cells). Secondary endpoints included ORR, PFS, and OS in all pts, as well as ORR in pts with PD-L1 + tumors.

#### 4.2.3. Tislelizumab in First-Line Setting

Two clinical trials are ongoing with the anti-PD1 tislelizumab in combination with first-line chemotherapy consisting of fluorouracil and platinum: the phase II single-arm BGB-A317-205 trial, also including gastric and gastro-esophageal neoplasms, and the randomized phase 3 BGB-A317-306, restricted only to ESCC.

### 4.3. Ongoing Trials in Localized Disease

Different studies are evaluating ICIs in combination with neoadjuvant and definitive chemoradiotherapy.

In particular, a phase II/III trial (NCT03604991) is investigating treatment with nivolumab alone or in combination with ipilimumab in addition to standard of care chemo-radiotherapy in patients with esophageal and GEJ adenocarcinoma who are undergoing surgery. The primary endpoint of the first part of the study is pathological complete response rate (pCRR), whereas the second part of the trial will evaluate DFS.

The upcoming KEYNOTE-975 is a phase III randomized trial evaluating pembrolizumab in combination with definitive chemo-radiotherapy with fluorouracil and oxaliplatin or cisplatin in patients with locally advanced ESCC, EAC, and Siewert I adenocarcinoma of GEJ. Another phase III trial, BGB-A317-311, is currently assessing the efficacy of tislelizumab vs. placebo in combination with concurrent chemoradiotherapy (with cisplatin and paclitaxel) in localized ESCC. Primary endpoints are OS and event-free survival defined as the time from randomization to local or distant recurrence or death.

Among the current studies in the adjuvant setting, CheckMate-577 is a phase III study of adjuvant nivolumab compared to placebo in patients with lower esophageal and EGJ cancer. Before randomization, patients must have completed preoperative CRT followed by surgery and have been diagnosed with residual pathological disease after radical complete resection. Primary endpoints are OS and DFS.

A brief description of selected ongoing trials in non-metastatic esophageal cancer is reported in Table 3.

## 5. Potential Molecular Selection

Results from the abovementioned trials suggest that only a subgroup of patients could derive a meaningful and long-term benefit from immunotherapy. For this reason, preclinical and translational studies are trying to identify putative predictive factors in order to select esophageal cancer patients more responsive to immune treatment.

### 5.1. PD-L1 Expression

PD-L1 expression has been studied in a magnitude of cancers—it has resulted in being predictive of response to immune checkpoint inhibitors among different cancer types [35,50,51]. Notably in the phase II trial KEYNOTE-059 exploring the efficacy of pembrolizumab as the third line of treatment of gastric cancer, an ORR of 22.7% vs. 16.4% was observed in a PD-L1-positive population compared to a negative one, although PD-L1-negative tumors still responded [52]. However, it is important to consider that results from KEYNOTE-059 led to the FDA approval for pembrolizumab in gastric or GEJ adenocarcinoma whose tumors express PD-L1.

In esophageal cancers, PD-L1 expression and its association with survival remain controversial—in order to address this issue, in 2018, Yu and Guo performed a systematic review and meta-analysis on a total of 3306 patients (18 published studies included) who underwent surgery. PD-L1 status, assessed by means of immunohistochemistry techniques on tumor cells, was found to have an unfavorable prognostic impact in terms of OS without significant correlations with DFS. The rate of PD-L1 over-expression ranged from 14.5% to 63.3%. They also performed a subgroup analysis according to histology—in patients with ESCC, the combined hazard ratio confirmed PD-L1 overexpression to be a poor prognostic index of OS. Due to the limited number of included studies involving EAC, no evidence about this histology could be stated [53]. In another series of 150 ESCC, the over-expression of PD-L1 and PD-L2 rated around 64% and 42%, respectively, with a significant correlation between the expression of both. At the multivariate analysis, high levels of PD-L1 were correlated with a worse DFS, together with an advanced pathological stage [54].

In KEYNOTE-180, CPS PD-L1≥10% was observed in 47.9% of patients and seemed to be associated with a slight improvement of tumor response compared to PD-L1 negativity [15]. On this basis, results from KEYNOTE-181 specified pembrolizumab as a new standard of care in esophageal cancer patients with a CPS PD-L1 ≥10% [20]. On the contrary, in CheckMate-032 gastro-esophageal cohort PD-L1 expression was assessable in 79% of cases; a sample was considered positive if it had ≥ 100 evaluable tumor cells and ≥ 1% PD-L1 staining of tumor cell membranes. The prevalence of PD-L1 positivity was 38%, 24%, and 30%, respectively, for each cohort N3, N1+I3, and N3+I1, and PD-L1 expression did not correlate with tumor response. To conclude, the role of PD-L1 both as a prognostic and predictive biomarker in esophageal cancer is still controversial [27].

### 5.2. Mismatch Repair Deficiency and DNA Damage Response

Mismatch repair deficient (dMMR) tumors harbor 10 to 100 times more mutations than mismatch repair proficient (pMMR) tumors—principally in the repetitive DNA sequences, called microsatellites—which for this reason have developed a very high instability (MSI-H). These tumors present a very high tumor mutational burden (TMB), which results in a large number of mutation-associated neo-antigens that might be recognized by the immune system [55]. In a pivotal phase II trial, the response to the anti-PD1 pembrolizumab was evaluated in 86 patients affected by MSI-h/dMMR heavily pre-treated metastatic disease of different cancer types including gastro-esophageal tumors. In this cohort, the microsatellite instability resulted in a strong predictor of response with an overall response rate of 53% [56]. Thanks to this study, pembrolizumab received in 2017 the first tissue/site-agnostic approval for treatment of patients with unresectable or metastatic MSI-H solid tumors that have progressed after all standard treatment options [57]. These very promising results have been recently confirmed in the phase II trial KEYNOTE-158 [58]. Despite these exciting data, we have to remind ourselves that the rate of MSI-H tumors in the esophageal disease is less than 2%, with an absence of MSI-H tumors in the EAC analyzed by The Cancer Genome Atlas (TGCA) [8,59,60]. Esophageal cancer has been initially granted as neoplasia harboring high TMB [61]. Recently, Parikh et al. conducted a comprehensive analysis of genes involved in DNA damage response (DDR) in 17,486 samples of patients affected by gastrointestinal neoplasms with the aim of characterizing DDR defection in this population and exploring its potential correlation with TMB with a view to the potential implication in immunotherapy. DDR alterations were found in a significant proportion of esophageal adenocarcinoma samples (467/2501, 19%), median TMB was 5.0 mut/Mb, and high TMB cases (defined as TMB ≥20 mut/Mb) were only 59 (2.4%) [60]. Further investigations based on these preliminary findings are warranted in order to better understand any therapeutic implication of this research line.

### 5.3. Role of the Gut Microbiota

In the last few years, the interest in the gut microbiome’s composition and its genetic polymorphisms as modulators of therapeutic response to ICI inhibitors had stepwise increase [62,63,64]. One of the first pieces of evidence of the role of microbiota derives from melanoma, where its composition seems to favor the response to anti-PD1 agents by increasing antigen presentation and improving T cell function [64]. Moreover, the dysbiosis transiently caused by antibiotic chemotherapy may represent a predictor of resistance to ICIs, and can be reversed with a re-sensibilization to immunotherapy by means of responders’ fecal microbiota transplantation or colonization with favorable commensals [63]. On the basis of this background, different studies have also focused on the esophageal microbiota, demonstrating differences among the microbiome of the normal epithelium (*Streptocuccous viridans* as a major component), of Barrett’s esophagus, of the EAC, and of the ESCC [65,66]. A global alteration of the microbiome in the distal esophagus is known in patients affected by reflux esophagitis and Barrett’s esophagus, with a switch to Gram-negative bacteria and a consequent production of a consistent amount of lipopolysaccharide, which may increase toll-like receptor 4 signaling and expression of downstream inflammatory cytokines leading to the progression from inflammation to adenocarcinoma development. Additionally, *Campylobacter* species, characterized by a potential pathogenetic role [67], are relevantly present both in Barrett’s esophagus and in adenocarcinoma, suggesting a possible role in tumor progression [66]. The framework in ESCC is less defined—the presence of Clostridiales and Erysipelotrichaceae in gastric microbiota seem associated with esophageal squamous dysplasia and ESCC [68]. Furthermore, the presence of *Fusobacterium nucleatum* seems to be associated with a worse prognosis of esophageal squamous cell carcinoma, probably due to its activation of chemokines [66]. In a future perspective, a better characterization of the esophageal microbiota, as well as the mechanisms involved in microbiota regulation, could be relevant in order to identify patients that could benefit from immunotherapy.

On this topic, increasing evidence has shown that the diet and especially polyunsaturated fatty acids (PUFAs) represent one of the strongest selective pressure for microbial communities within the gastrointestinal tract. In murine models, high levels of omega-6 PUFAs cause a reduction of microbiota richness, whereas omega-3 PUFAs seem to favorably modify the gut microbiota [69,70] and directly activate tumor-killing cytokines, overcoming the tumor-related immunosuppression. Due to the linkage between the immune system, inflammation, and gut microbiome, many authors hypothesize that long-chain PUFAs could represent a biomarker for patients’ healthy gut microbiota and a potential therapeutic agent modulating the composition of the microbiome and subsequently the response to immunotherapy [70].

For the abovementioned reasons and the constitution of the esophageal microbiota, PUFAs could be investigated as a future potential regulator of the cross-talk between gut microbiome and ICI treatment.

## 6. Conclusions

To conclude, emerging data of recently presented pivotal trials seem to support the strong preclinical rationale of a potential role of immunotherapy in the therapeutic armamentarium of esophageal cancer both in advanced and in localized disease. In view of the already available results of phase I/II studies, ongoing phase III trials are awaited in the metastatic setting with the purpose of understanding if checkpoint inhibitors alone or in combination with other immune agents or chemotherapy could become a new standard already from the first line of treatment. At the same time, the potential synergistic effect of combining radiotherapy, chemotherapy, and immune treatment is currently explored in localized disease. The availability of data from these ongoing trials will help us to figure out the optimal positioning of immunotherapy in the multimodality treatment of esophageal cancer. Finally, results from clinical trials should necessarily be accompanied by a better understanding of molecular mechanisms that underlie tumor response to immunotherapy in order to identify predictive biomarkers that enable the selection of patients for optimal treatment.

## Figures and Tables

**Table 1 ijms-21-01658-t001:** Summary of selected presented trials in esophageal cancer.

Clinical Trial Identifier	Setting/Line	Line	Phase	Site and Histology	Treatment Arm(s)	Accrual	Primary Endpoint	Results
ONO-4538-07/JapicCTI-No. 142422 ATTRACTION-01	Metastatic	≥2	II	ESCC	Nivolumab 3 mg/kg q2w	65	ORR	ORR with RECIST 17% (CI 95% 10–28); DCR 42% (CI 95% 31-54); 26% grade ≥3 AEs
NCT02559687 KEYNOTE-180	Metastatic	≥2	II	ESCC or esophageal/GEJ adenocarcinoma	Pembrolizumab 200 mg q3w	121	ORR	ORR in ITT: 9.9% (CI 95% 5.2–16.7); in PD-L1 positive (CPS ≥ 10%): 13.8% (CI 95% 6.1–25.4); in PD-L1 negative: 6.3% (CI 95% 1.8–15.5)
NCT01928394 CheckMate-032	Metastatic	≥2	I/II	Esophagogastric adenocarcinoma	Nivo 3 mg/kg q2w vs. Nivo 1 mg/kg + Ipi 3 mg/kg q3w vs. Nivo 3 mg/kg + Ipi 1 mg/kg q3w	160	ORR	ORR in Nivo group: 12% (CI 95% 5–23); in Nivo 1 + Ipi 3: 24% (CI 95% 13–39); in Nivo 3 + Ipi 1: 8% (CI 95% 2–19); 17% vs 47% vs 27% respectively grade ≥3 trAEs
NCT02054806 KEYNOTE-028	Metastatic	≥2	Ib	ESCC or esophageal/GEJ adenocarcinoma PD-L1 positive (≥1%)	Pembrolizumab 10 mg/kg q2w	23	Safety, ORR	39% trAEs (rash, decreased appetite, decreased lymphocyte count); no treatment-related death; 26% SIAEs (hypotyroidism, enterocolitis, hypertyroidism, adrenal insufficiency, generalized rash); ORR 30% (CI 95% 13–53)
NCT02915432 JS001-Ib-CRP-1.0	Metastatic	2	Ib/II	ESCC	JS001 3 mg/kg q2w	56	Safety; activity (evaluated in 34 pt)	trAEs grade 1–2 only; ORR 23.5%; DCR 64.7%
NCT02569242 ATTRACTION-3	Metastatic	2	III	ESCC	Nivo 240 mg q2w vs CT (TXT or PTX)	419	OS	median 10.9 vs 8.4 months (HR 0.77, 95% CI 0.62–0.96; *p =* 0.019)
NCT02564263 KEYNOTE-181	Metastatic	2	III	ESCC or EAC	Pembro 200 mg q3w vs CT (TXT, PTX, or CPT-11)	628	OS in ITT group OS in ESCC group OS in PD-L1 positive (CPS ≥ 10%) group	In ITT: median 7.1 vs 7.1 months (HR 0.89, CI 95% 0.75-1.05, *p* = 0.0560) In ESCC: median 8.2 vs 7.1 months (HR 0.78, CI 95% 0.63-0.96, *p* = 0.0095) In PD-L1-positive: median 9.3 vs 6.7 months (HR 0.69; CI 95% 0.52-0.93, *p* = 0.0074)
NCT02658214 D419SC00001	Metastastic	1	Ib	ESCC	Durva 1.5 g + Treme 75 mg q4w + 5FU + CDDP	6	Safety and tolerability, ORR	100% trAE; 50% grade ≥3 trAEs (only neutropenia, due to 5FU + CDDP); 30% grade 1 or 2 immune-mediated AEs (diarrhea, pruritus, rash, and increased AST). ORR: 30%
NCT03469557 BGB-A317-205	Metastastic	1	II	ESCC	Tislelizumab 200 mg q3w + CDDP+5FU	15	Safety and tolerability, ORR	1 grade 5 trAE (hepatic dysfunction); 4 discontinuations due to AEs (grade 3 tracheal fistula, lung infection, increase sGOT/sGPT, grade 2 pneumonitis). Activity data unmatured
NCT02639065 BTCRC-ESO14-012	Resected post-CCRT and residual disease	Adj	II	Esophageal and GEJ adenocarcinoma	Durvalumab	24	RFS	1-year RFS: 79.2%

**Abbreviations:** AE: adverse event; CDDP: cisplatin; CI: confidence interval; CPS: combined positive score; CPT-11: irinotecan; CRT: chemoradiotherapy; CT: chemotherapy; DCR: disease control rate; DoR: duration of response; Durva: durvalumab; EAC: esophageal adenocarcinoma; ESCC: esophageal squamous cell cancer; GC: gastric cancer; GEJ: gastro-esophageal junction; HR: hazard ratio; Ipi: ipilimumab; ITT: intention-to-treat; NCT: number of clinical trial (https://clinicaltrials.gov/); Nivo: nivolumab; PD-L1: programmed death-ligand 1; Pembro: pembrolizumab; PFS: progression-free survival; q2w: every two weeks; ORR: objective response rate; OS: overall survival; PTX: paclitaxel; RECIST: response evaluation criteria in solid tumors; RFS: relapse-free survival; sGOT: serum glutamic oxaloacetic transaminase; sGPT: serum glutamic pyruvic transaminase; trAE: treatment-related adverse event; Treme: tremelimumab; TXT: docetaxel; 5FU: 5-fluorouracil.

**Table 2 ijms-21-01658-t002:** Selected ongoing trials in metastatic esophageal cancer.

Clinical Trial Identifier	Setting	Line	Phase	Site and Histology	Treatment Arm(s)	Primary Endpoint	Recruiting	Target Accrual
NCT03143153 CheckMate-648	Metastatic	1	III	ESCC	Nivo + Ipi vs. Nivo + CT (5FU+CDDP) vs. CT	OS, PFS in PD-L1 ≥ 1%	Active, not recruiting	939
NCT03189719 KEYNOTE-590	Metastatic	1	III	EAC, ESCC, adenocarcinoma of GEJ (Siewert type 1)	Pembro + CT (5FU+CDDP) vs. placebo + CT	OS, PFS in ITT and in PD-L1 (CPS ≥10%)	Active, not recruiting	700
NCT03783442 BGB-A317-306	Metastatic	1	III	ESCC	Tisle + CT (Platinum+5FU/PTX/cape) vs. placebo + CT	PFS, OS	Recruiting	480
NCT03469557 BGB-A317-205	Metastatic	1	II	ESCC and GC/GEJ carcinoma	ESCC: Tisle+ CT (CDDP+5FU) GC/GEJ carcinoma: Tisle + CT (LOHP+cape)	ORR, DoR, DCR, PFS, pharmacokinetic valuations, host immunogenicity	Active, not recruiting	30
NCT03430843 BGB-A317-302	Metastatic	2	III	ESCC	Tisle vs. CT (PTX/TXT/CPT-11)	OS	Recruiting	450
NCT03416244 RAMONA trial	Metastatic elderly (>65 years)	2	II	ESCC	Nivo + Ipi vs. Nivo	OS	Recruiting	75
NCT03811379 PD-1/SCCE	Metastatic	≥2	II	SCEC	Toripalimab	ORR	Recruiting	43
NCT03544736 INEC-study	Cohort A: advanced	Cohort A: palliative	I/II	EC/GEJ carcinoma	Cohort A: Nivo + palliative RT	Safety	Recruiting	54
NCT02735239 LUD2015-005	Cohorts A and B: metastatic	Cohorts A and B: 1	I/II	Esophageal and GEJ carcinoma	Cohorts A and B: Durva +/- Treme → Durva +/- Treme + FOLFOX	DLT, safety	Recruiting	75

**Abbreviations:** cape: capecitabine; cCRR: clinical complete response rate; CDDP: cisplatin; CPS: combined positive score; CPT-11: irinotecan; CT: chemotherapy; d: day; DCR: disease control rate; DLT: dose-limiting toxicity; DoR: duration of response; Durva: durvalumab; EAC: esophageal adenocarcinoma; EC: esophageal cancer; ESCC: esophageal squamous cell cancer; GC: gastric cancer; GEJ: esophagogastric junction; Ipi: ipilimumab; ITT: intention-to-treat; LOHP: oxaliplatin; NCT: number of clinical trial (https://clinicaltrials.gov/); Nivo: nivolumab; ORR: objective response rate; OS: overall survival; Pembro: pembrolizumab; PFS: progression free survival; PTX: paclitaxel; RFS: relapse free survival rates; SCeE: small cell esophageal cancer; TXT: docetaxel; Treme: tremelimumab; 5FU: 5-fluorouracil.

**Table 3 ijms-21-01658-t003:** Selected ongoing trials in non-metastatic esophageal cancer.

Clinical Trial Identifier	Setting	Line	Phase	Site and Histology	Treatment Arm(s)	Primary Endpoint	Recruiting	Target Accrual
NCT03777813 PRODIGE 67-UCGI33 ARION Trial	Locally advanced unresectable	dCCRT	II	EC	Durvalumab + dCCRT (RT + 5FU+ LOHP) vs. dCCRT	PFS	Recruiting	120
NCT04005170 TORIDEFEC	Locally advanced unresectable	dCCRT	II	ESCC	Toripalimab + dCCRT (RT + CDDP + PTX)	cCRR	Recruiting	42
NCT04084158	Locally advanced unresectable	dCCRT	II	ESCC	Triprizumab + dCCRT (RT + CBDCA + taxane) vs. dCCRT	PFS	Recruiting	100
NCT03437200 EORTC-1714	Early stage or locally advanced unresectable	dCCRT	II	EC	Nivo + dCCRT (RT+ 5FU + LOHP) vs. Nivo + Ipi + dCCRT	PFS	Recruiting	130
NCT03985670 HenanCH immunotherapy001	Locally advanced resectable	NeoAdj	II	ESCC	Toripalimab d1 + PTX d1 + CDDP d1 vs. PTX d1 + CDDP d1 + Toripalimab d3	pCRR	Recruiting	30
NCT04006041 TORINEOEC	Locally advanced resectable	NeoAdj	II	ESCC	Toripalimab + CCRT (RT + PTX + CDDP)	pCRR	Recruiting	44
NCT03604991	Locally advanced resectable	Perioperative	II/III	Esophageal and GEJ adenocarcinoma (Siewert I and II)	Neoadj CCRT (RT + CBDCA + PTX) vs. Neoadj CCRT + Nivo vs. Nivo vs Nivo + Ipi	pCRR DFS	Recruiting	278
NCT03288350	Locally advanced resectable	Perioperative	II	Esophageal/GEJ/gastric adenocarcinoma	Perioperative mDCF + Avelumab	pCRR	Recruiting	55
NCT03957590 BGB-A317-311	Early stage	dCCRT	III	ESCC	Tisle + dCCRT (RT + CDDP + PTX) vs. Placebo + dCCRT	PFS	Recruiting	316
NCT03784326	Locally advanced resectable	NeoAdj	I	Esophageal and GEJ adenocarcinoma (Siewert I and II)	Atezolizumab + 5FU + LOHP	pCRR	Recruiting	30
NCT 04210115 KEYNOTE-975	Locally advanced unresectable	dCCRT	III	EC + GEJ adenocarcinoma (Siewert I)	Pembro + dCCRT (RT + 5FU + LOHP/CDDP) → Pembro vs. placebo + dCCRT (RT + 5FU + LOHP/CDDP) → placebo	OS/EFS	Not yet recruiting	600
NCT02743494 CheckMate-577	Resected post-CCRT	Adj	III	Esophageal and GEJ carcinoma	Nivo vs. placebo	DFS	Active, not recruiting	760
NCT03044613	Locally advanced resectable	NeoAdj	Ib	EC/GEJ carcinoma	Cohort A: Nivo → Nivo + CCRT (RT + CBDCA + PTX) Cohort B: Nivo + Relatlimab → Nivo + Relatlimab + CCRT	Safety	Recruiting	25
NCT03544736 INEC-study	Cohort B: locally advanced unresectable Cohort C: resectable	Cohort B: dCCRT Cohort C: NeoAdj CCRT	I/II	EC/GEJ carcinoma	Cohort B: Nivo + dCCRT (RT + CBDCA + PTX) → Nivo Cohort C: Nivo + CCRT (RT + CBDCA + PTX) → surgery → Nivo	Safety	Recruiting	54
NCT02735239 LUD2015-005	Cohorts C and D: locally advanced	C: NeoAdj C-FLOT: perioperative D: NeoAdj	I/II	Esophageal and GEJ carcinoma	Cohort C: Durva → Durva + Cape + LOHP → surgery Cohort C-FLOT: Durva → Durva + FLOT → surgery Cohort D: Durva + → Durva + CCRT (RT+ CBDCA + PTX) → surgery	DLT, safety	Recruiting	75

**Abbreviations:** Cape: capecitabine; CBDCA: carboplatin; cCRR: clinical complete response rate; CCRT: concomitant chemo-radiotherapy; CDDP: cisplatin; CT: chemotherapy; d: day; dCCRT: definitive concurrent chemoradiotherapy; DCR: disease control rate; DLT: dose-limiting toxicity; Durva: durvalumab; EC: esophageal cancer; EFS: event free survival; ESCC: esophageal squamous cell cancer; GC: gastric cancer; GEJ: esophagogastric junction; Ipi: ipilimumab; LOHP: oxaliplatin; NCT: number of clinical trial (https://clinicaltrials.gov/); NeoAdj: neoadjuvant; Nivo: nivolumab; 5FU: 5-fluorouracil; ORR: objective response rate; OS: overall survival; pCRR: pathological complete response rate; Pembro: pembrolizumab; PFS: progression-free survival; PTX: paclitaxel; RFS: relapse-free survival rates.

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
