# Peer review of "Immune Checkpoint Inhibitors in Esophageal Cancers: Are We Finally Finding the Right Path in the Mist?"

_ijms, 2020, doi:10.3390/ijms21051658_

Round 1

Reviewer 1 Report

The authors presented a comprehensive review on the use of immune checkpoint inhibitors (ICI) against oesophageal cancer.  They covered a variety of trials to great detail and offered some molecular markers that are related to the disease. However, since part of the review’s aim is to connect the disease with promising biomarkers and future perspectives, the manuscript can be improved by adding the citation of recent studies especially on the role of the microbiome.

One of the recent developments on the use of ICI are the findings that its efficacy depends on the patients’ gut microbiome. Please refer to references below.

Gong, J et al (2018). Development of PD-1 and PD-L1 inhibitors as a form of cancer immunotherapy: a comprehensive review of registration trials and future considerations. J Immunother Cancer 6: 8, doi: 10.1186/s40425-018-0316-z. Routy, B et al. (2018). Gut microbiome influences efficacy of PD-1-based immunotherapy against epithelial tumors. Science 359:91-97,10.1126/science.aan3706. Gopalakrishnan, V et al. (2018). Gut microbiome modulates response to anti-PD-1 immunotherapy in melanoma patients. Science 359:97-103. doi: 10.1126/science.aan4236.

It is worth citing the reference below showing the relationship between the microbiome and oesophageal cancer which bridges the microbiome and ICI in oesophageal cancer

Baba Y et al. (2017) Review of the gut microbiome and esophageal cancer: Pathogenesis and potential clinical implications. Ann Gastroenterol Surg. 1:99–104. DOI: 10.1002/ags3.12014

Furthermore, the authors should consider the following references where PUFAs have been shown to directly influence the microbiome and provides insight on potential mechanisms of action, especially with respect to inflammation that may be relevant to cancer in general which may have implications linking the microbiome and ICI.  The articles below provide additional mechanisms on how PUFAs can regulate the microbiome and ICI which will strengthen the manuscript with additional biomarkers to consider for oesophageal cancer.

Park JM et al (2013) Omega-3 Polyunsaturated Fatty Acids as Potential Chemopreventive Agent for Gastrointestinal Cancer. J Cancer Prev 18:201-208. Eltweri et al. (2018) Effects of Omegaven®, EPA, DHA and oxaliplatin on oesophageal adenocarcinoma cell lines growth, cytokine and cell signal biomarkers expression. Lipids in Health and Disease 17:19. DOI 10.1186/s12944-018-0664-1 Ilag LL (2018) Are Long-Chain Polyunsaturated Fatty Acids the Link between the Immune System and the Microbiome towards Modulating Cancer? Medicines 2018, 5(3), 102 doi.org/10.3390/medicines5030102 Constantini L et al (2017) Impact of Omega-3 Fatty Acids on the Gut Microbiota. Int J Mol Sci.2017 Dec 7;18(12). pii: E2645. doi: 10.3390/ijms18122645 On the text, some of the sections are too long and makes it difficult to read. I suggest creating sub-sections to provide more clarity and make it easier for the reader to browse through the article. Section 2.1 From initial phase 1-2 randomized phase 3 trials. I suggest creating subsections for each of the trials such as 2.1.1 Keynote-028 2.1.2 ATTRACTION 1 2.1.3 KEYNOTE-181…… and so on Section 2.2 I suggest creating subsections for the different antibodies such as 2.2.1 Nivolumab and Ipilimumab combination 2.2.2 M7824 – bifunctional fusion receptor protein… Toripalimab…… and so on

Author Response

Response 1. We thank the reviewer and we agree with his suggestion. We updated chapter 4 adding a section entitled  The role of gut microbiota (section 4.3) summarizing the evidences regarding modulation of immunotherapy response by microbiota, the characteristics of microbiota in relation to esophageal cancer and the potential modulating role of FUFAs.

Response 2. We thank the Reviewer for his suggestion. We created subsections in order to make reading easier.

Reviewer 2 Report

The authors reviewed the current situation and the future perspective of immune checkpoint inhibitors for esophageal cancer. The results of recent clinical trials on ICI for esophageal cancer are promising, and this manuscript nicely summarized the results.

I think this manuscript is well-written and and informative. It will provide useful clinical information for readers.

There are several grammatical errors that should be corrected.

Author Response

Response 1. We thank the reviewer for his comment. We provided the required English changes.